# Chelator Iminodisuccinic Acid Regulates Reactive Oxygen Species Accumulation and Improves Maize (*Zea mays* L.) Seed Germination under Pb Stress

**DOI:** 10.3390/plants11192487

**Published:** 2022-09-22

**Authors:** Yifei Zhang, Yishan Sun, Weiqing Li, Jiayu Li, Rongqiong Xu, Jiarui Du, Zesong Li, Guibin Li, Kejun Yang

**Affiliations:** 1Heilongjiang Provincial Key Laboratory of Modern Agricultural Cultivation and Crop Germplasm Improvement, Department of Agronomy, Heilongjiang Bayi Agricultural University, Daqing 163319, China; 2Key Laboratory of Low-Carbon Green Agriculture in Northeastern China, Ministry of Agriculture and Rural Affairs, Daqing 163319, China

**Keywords:** maize, chelator, lead stress, ROS, hydrogen peroxide, NADPH oxidase, seed germination

## Abstract

To explore the effects of iminodisuccinic acid (a chelating agent) on maize (*Zea mays* L.) seed germination under lead (Pb) stress, we comparatively analyzed the effects of applying different concentrations of iminodisuccinic acid (0, 5, 20, and 100 mmol·dm^−3^) and combined an addition of exogenous substances regulating reactive oxygen species production on maize seed germination, seedling growth, H_2_O_2_ content, NADPH oxidase activity, and antioxidant enzyme activities under Pb-stressed and Pb-free conditions. Iminodisuccinic acid (100 mmol·dm^−3^) significantly delayed seed germination under normal germination conditions and alleviated the inhibitory effects of Pb stress (20 mmol·dm^−3^) on seed germination. Under normal conditions (without Pb stress), the iminodisuccinic acid-induced inhibition of seed germination was enhanced by treatment with dimethylthiourea (a specific scavenger of reactive oxygen species) or diphenyleneiodonium chloride (a specific inhibitor of NADPH oxidase), but diminished by treatment with H_2_O_2_, CaCl_2_, diethyldithiocarbamic acid (a specific inhibitor of superoxide dismutase), or aminotriazole (a specific inhibitor of catalase). Under Pb stress, iminodisuccinic acid partially eliminated the excessive H_2_O_2_ accumulation, improved superoxide dismutase and catalase activity, and weakened the high NADPH oxidase activity. In addition, Ca^2+^ chelation may be essential for maintaining the reactive oxygen species’ balance and improving seed germination and seedling growth by iminodisuccinic acid supplementation in maize under Pb stress. The proposed iminodisuccinic acid supplementation-based method improved maize seed germination in Pb-polluted soil.

## 1. Introduction

In recent decades, rapid increases in urbanization and industrialization have caused the excessive release of heavy metals in farmlands, which has led to ecosystem damage [1,2,3]. Heavy metal accumulation in soils is of great concern in agricultural production due to its adverse effects on food safety and marketability, crop growth due to phytotoxicity, and the environmental health of soil organisms [4,5,6,7,8]. Industrial wastewater and urban sewage have been increasingly used in urban and suburban agricultural irrigation systems to partially resolve the problem of wastewater treatment [9,10]. Although the concentration of heavy metals in industrial wastewater and urban sewage is usually low [11], the long-term usage of these wastewater sources to irrigate agricultural land culminates in the enrichment of and significant increase in heavy metals in soil [12]. Additionally, the global oil and gas exploitation industry plays an essential, indispensable, and dynamic role in the international energy matrix [13]. Drilling associated with oil and gas exploitation projects can generate significant quantities of waste [14] that are extremely difficult to manage due to the presence of several heavy metal pollutants [15,16]. These heavy metal pollutants also exert long-term ecological effects on populations and communities, and thus, pose a threat to biodiversity and agricultural ecosystem integrity [17,18].

Studies have shown that using high-accumulating crops (metal accumulators), such as nonfood products and dedicated bioenergy crops, may be a feasible strategy for the phytoremediation of heavy metal-polluted soils [19,20]. Among diverse crop plants, maize (*Zea mays* L.) exhibits the advantages of high biomass production and metal tolerant capacity; thus, it has been widely adopted for the phytomanagement of low- and medium-grade metal-contaminated soils [21,22,23]. Moreover, with the increasing global demand for maize due to population growth and industrial expansion, phytoremediation techniques that use maize to decontaminate contaminated environments represent cost-effective, environmentally friendly, and socially acceptable methods for remediating metal-contaminated soils [24,25,26,27]. However, seed germination and seedling growth are critical stages in maize plant development, and the presence of heavy metals may impair growth or even render plant formation non-viable [28]. Therefore, in a phytoremediation program, successful germination and seedling development (the initial stages of plant growth) are essential for obtaining maize plants capable of extracting heavy metal contaminants from the soil [29].

Lead (Pb) is a highly toxic and commonly occurring heavy metal contaminant in the environment [30,31]. Of the various heavy metals, Pb is considered to be the second most toxic after arsenic [32]. Pb toxicity inhibits seed germination and plant growth by negatively affecting the physiological traits of plants [33,34,35]. Studies have shown that reactive oxygen species (ROS) are central components of plant adaptations to biotic and abiotic stresses and function as signaling molecules to either positively or negatively regulate seed germination [36,37,38]. In seeds, ROS play roles in endosperm weakening, mobilization of seed reserves, protection against pathogens, and programmed cell death [39]. In addition, neither high nor low levels of ROS are conducive to seed germination [40]. Research has established that Pb induces ROS production and antioxidant defense responses. These responses include changes in the activities of ROS scavenging enzymes (e.g., superoxide dismutase (SOD) and catalase (CAT)) and NADPH oxidase (NOX; the key enzyme producing ROS) in plants [41,42,43,44,45].

Direct seeding technology is widely used in mechanized maize production [46,47]. Phytoremediation assisted by plant growth regulators is a new, promising technique for improving maize seed germination, seedling growth, and population formation. This technique also increases grain yield and improves the efficiency and applicability of phytoremediation practices [48,49]. Chelating agents, such as ethylenediaminetetraacetic acid (EDTA) and iminodisuccinic acid (IDS), have been widely used to enhance soil washing and metal phytoextraction. Different from the poor biodegradability of EDTA, a major proportion of IDS biodegrades after one week of treatment [50]. Moreover, IDS is characterized by its low environmental impact owing to its low toxicity [51]. However, the effects of IDS on maize growth under lead (Pb) stress remain unclear. 

Hence, the objective of this study was twofold: to analyze the effects of IDS on seed germination and seedling growth under Pb-stressed and Pb-free conditions and to explore the regulatory mechanism underlying the effects of IDS on the seed germination process in terms of ROS balance to provide theoretical and technical support to facilitate the efficient implementation of phytoremediation projects for contaminated soil treatment.

## 2. Results

### 2.1. Effects of IDS and Pb on Maize Seed Germination

In the absence of heavy metal stress (Figure 1), an increase in the concentration of IDS gradually delayed the germination of maize seeds. Treatment with 100 mmol·dm^−3^ IDS significantly delayed seed germination. Compared with the germination rate (GR) of seeds in the distilled water (control) treatment group, the GR of seeds treated with 100 mmol·dm^−3^ IDS was approximately 58%, 62%, 52%, and 9% less at 1, 2, 3, and 4 d after sowing, respectively (Figure 1, *p* < 0.05). However, 5 d after sowing, significant differences were not observed in the final GR of seeds across the IDS treatment groups.

Although the final GR of seeds was reduced to approximately 18% under heavy metal stress (20 mmol·dm^−3^ PbCl_2_), the IDS treatment effectively alleviated the inhibitory effect of Pb stress on seed germination in maize (Figure 2). For instance, compared with the GR after the treatment with Pb alone, the GR after the treatment with 100 mmol·dm^−3^ IDS under conditions of Pb stress was approximately 2.03-, 1.85-, 1.86-, 1.92-, and 1.85-fold higher at 1, 2, 3, 4, and 5 d after sowing, respectively (Figure 2, *p* < 0.05). Similar changes were observed for maize seed germination under Pb stress in groups treated with 5 and 20 mmol·dm^−3^ IDS (Figure 2, *p* < 0.05).

### 2.2. Effects of Pb and IDS on Seedling Growth

Five days after sowing, we assessed the growth of maize seedlings treated with different concentrations of IDS under Pb-stressed and Pb-free conditions. Under normal incubation conditions, treatment with high concentrations of IDS slightly but significantly reduced the growth of maize seedlings. For instance, the seedling sprout length decreased by approximately 7% and 12% after treatment with 20 and 100 mmol·dm^−3^ IDS, respectively (Figure 3, *p* < 0.05). By contrast, Pb stress-inhibited seedling growth was significantly alleviated by IDS. Compared with the sprout length of seedlings under Pb stress alone, that of seedlings treated with 5, 20, and 100 mmol·dm^−3^ IDS while under Pb stress was approximately 20%, 50%, and 67% higher, respectively (Figure 3, *p* < 0.05).

### 2.3. Effects of ROS-Related Reagents on Seed Germination

The germination of maize seeds treated with Pb, IDS, or their combination was significantly affected by treatment with redox reagents, such as hydrogen peroxide (H_2_O_2_) and dimethylthiourea (DMTU) (Figure 4). After the maize seeds were incubated with Pb or Pb+IDS for 24 h, the GR of seeds treated with H_2_O_2_ was significantly reduced by approximately 60% and 27%, respectively, compared with seeds without H_2_O_2_ treatment (Figure 4, *p* < 0.05). A contrasting effect was observed in seeds treated with DMTU. For instance, in seeds incubated with Pb and Pb+IDS for 24 h, treatment with DMTU increased the GR by approximately 80% and 22%, respectively (Figure 4, *p* < 0.05). However, the GR of IDS-treated seeds increased by 46% and decreased by 42% after treatment with H_2_O_2_ and DMTU, respectively (Figure 4, *p* < 0.05).

We investigated the effects of diphenyleneiodonium chloride (DPI) (a specific inhibitor of NOX) and CaCl_2_ on maize seed germination in the distilled water (control), Pb, IDS, and Pb+IDS treatment groups during the first 24 h of incubation (Figure 5). In the control and IDS treatment groups, treatment with DPI reduced the GR of maize seeds by approximately 31% and 50%, respectively, compared with that in the DPI- and CaCl_2_-free conditions (Figure 5, *p* < 0.05). However, after the addition of DPI, the GR was approximately 120% and 26% higher in the Pb and Pb+IDS treatment groups, respectively, than in the corresponding DPI- and CaCl_2_-free condition (Figure 5, *p* < 0.05). Furthermore, compared with that in the DPI- and CaCl_2_-free conditions, the addition of CaCl_2_ increased the GR by approximately 21% and 25% in the control and IDS treatment groups, and reduced the GR by approximately 43% and 22% in the Pb and Pb+IDS treatment groups, respectively (Figure 5, *p* < 0.05).

Next, we investigated the effects of the antioxidant enzyme inhibitors diethyldithiocarbamic acid (DDC) (a specific inhibitor of SOD) and aminotriazole (ATZ) (a specific inhibitor of CAT) on the germination of maize seeds treated with Pb, IDS, or Pb+IDS. Twenty-four hours after the addition of DDC and ATZ, the GR of distilled water (control)- and IDS-treated maize seeds was significantly increased by approximately 16% and 33% (DDC) and by 20% and 44% (ATZ), respectively (Figure 6). However, treatment with DDC and ATZ enhanced the inhibitory effects of Pb and Pb+IDS on the germination of maize seeds. For instance, after treatment with DDC and ATZ, the GR of maize seeds was 30% and 24%, respectively, less than that of the DDC and ATZ-free seeds in the Pb+IDS-treated group (Figure 6, *p* < 0.05). 

### 2.4. Effects of Pb and IDS on H_2_O_2_ Content and NOX Activity

We determined the effects of Pb, IDS, and Pb+IDS on H_2_O_2_ accumulation in maize seeds after incubation for 1, 3, and 5 d (Figure 7). Throughout germination, the H_2_O_2_ content of maize seeds showed a significant increasing trend. In the control group treated with distilled water, the H_2_O_2_ content increased significantly by 53% and 94% in maize seeds after incubation for 3 and 5 d, respectively (Figure 7, *p* < 0.05), compared with that after incubation for 1 d. However, H_2_O_2_ accumulation was significantly affected by treatment with IDS and Pb. After incubation for 1 d, Pb and Pb+IDS increased the H_2_O_2_ content of germinating seeds by approximately 80% and 49%, respectively, compared with the distilled water (control) treatment (Figure 7, *p* < 0.05). After 1, 3, and 5 d of treatment with IDS, the H_2_O_2_ content of germinating seeds was approximately 29%, 22%, and 11% lower, respectively, than that in the distilled water (control) treatment group (Figure 7, *p* < 0.05).

Germination can significantly enhance NOX activity in maize seeds after incubation. Compared with the NOX activity after incubation for 1 d, germination increased the NOX activity by 47% and 79% after incubation for 3 and 5 d, respectively (Figure 8). After incubation for 3 d, Pb and Pb+IDS significantly enhanced NOX activity by approximately 85% and 46%, respectively, compared with that in maize seeds incubated with distilled water (control) (Figure 8, *p* < 0.05). However, IDS significantly decreased NOX activity by 33%, 25%, and 11% after germination for 1, 3, and 5 d, respectively, compared with that in the control group (Figure 8, *p* < 0.05).

### 2.5. Effects of Pb and IDS on Antioxidant Enzyme Activities

The effects of Pb, IDS, and Pb+IDS on the SOD and CAT activities in maize seeds after incubation for 1, 3, and 5 d are illustrated in Figure 9 and Figure 10. Germination of distilled water-treated maize seeds for 5 d enhanced the activities of SOD and CAT by approximately 53% and 72%, respectively, compared with that of seeds germinated for 1 d (Figure 9 and Figure 10, *p* < 0.05). After treatment with Pb and Pb+IDS, the SOD activity was increased and was approximately 20–30% and 40–56% higher, respectively, than that of the distilled water (control) treatment after incubation for 1–5 d (Figure 9, *p* < 0.05). Similar patterns of change were observed for CAT activity in maize seeds (Figure 10, *p* < 0.05). Compared with distilled water treatment, Pb and Pb+IDS treatment increased CAT activity in maize seeds incubated for 1–5 d, by approximately 10–17% and 18–41%, respectively (Figure 10, *p* < 0.05). However, IDS decreased SOD and CAT activities. Treatment with IDS reduced SOD and CAT activities by approximately 13–32% and 10–25%, respectively, in maize seeds after incubation for 1–5 d, compared with those in the control (Figure 9 and Figure 10, *p* < 0.05).

## 3. Discussion

To counter the significant increase in the Pb content of cultivated soils near industrial areas [52,53], chelating agents, such as EDTA and IDS, are used for phytoremediation projects because these agents can remove heavy metals from contaminated soils [43,54]. Maize is considered a potential phytoremediation crop for heavy metal-polluted soil [19,21,22,23]. Seen as direct seeding technology is commonly used in maize cultivation [46,47], seed germination and subsequent seedling growth are sensitive to the environment, especially in heavy metal-contaminated soils [55,56]. However, few studies have investigated the effects of chelators on seed germination and subsequent seedling growth. Therefore, in this study, we investigated the effects of IDS on maize seed germination and subsequent seedling growth under Pb-stressed and Pb-free conditions.

Although significant differences were not observed in the final GR, treatment with 100 mmol·dm^−3^ IDS noticeably delayed the germination of maize seeds that were not subjected to Pb stress (Figure 1). Similarly, higher concentrations of IDS also delayed or slightly (but significantly) inhibited the growth of maize seedlings (Figure 3). One plausible explanation for this phenomenon is that these chelators can chelate essential elements (such as Ca, Fe, Cu, Mn, and Zn) that are indispensable for seed germination [57]. These findings suggest that IDS, at least at low concentrations, exhibits low levels of adverse effects on seed germination and subsequent seedling growth in maize. However, under conditions of Pb stress, treatment with 5−100 mmol·dm^−3^ IDS significantly improved seed germination and seedling growth to different extents (Figure 2 and Figure 3). This finding indicates that treatment with Pb+IDS substantially reduces the toxic effects of Pb on seed germination and seedling growth. Based on the “oxidative window for germination” theory, low or high ROS levels deter seed germination [58]. Therefore, we speculated that ROS levels may be associated with the IDS-mediated alteration of redox balance during seed germination in maize.

Studies have suggested that ROS levels can be regulated by ROS producers (e.g., NOX) and ROS-scavenging enzymes (e.g., SOD and CAT) [41]. In this study, we investigated the effects of IDS on the GR of maize seeds by adding ROS-related redox agents under Pb-stressed and Pb-free conditions. The Pb treatment inhibited seed germination more than the distilled water treatment in the control group. Moreover, these inhibitory effects were aggravated by H_2_O_2_ but alleviated by DMTU (a specific scavenger of ROS) [59] after incubation for 24 h (Figure 4). This result suggests that Pb can induce ROS overproduction, which inhibits germination in maize seeds after incubation. IDS-delayed seed germination was improved by H_2_O_2_ and aggravated by DMTU in Pb-free conditions, and the opposite results were observed in Pb-stressed conditions (Figure 4). Therefore, under favorable growth conditions, IDS may delay maize seed germination and inhibit seedling growth by limiting the production of H_2_O_2_. Although adding H_2_O_2_ to the Pb+IDS treatment group inhibited the germination process, adding DMTU promoted it. Taken together, these results demonstrate that IDS positively regulates seed germination in maize by reducing the overproduction of H_2_O_2_ under conditions of Pb stress.

We also investigated the effects of DPI (a specific inhibitor of NOX) [60] and Ca^2+^ (a non-specific activator of NOX) [61] on seed germination in maize (Figure 5). The groups treated with Pb and Pb+IDS had a lower GR than the control group; however, the GR of the Pb+IDS treatment group was significantly higher than that of the Pb treatment group. The inhibition of seed germination in the Pb and Pb+IDS treatment groups was partially decreased by the DPI treatment but aggravated further by the CaCl_2_ treatment. Under Pb-free conditions, however, the IDS-delayed seed germination was enhanced by DPI but reversed by CaCl_2_. This phenomenon suggests that NOX-mediated ROS play a key role in IDS-regulated seed germination under Pb-stressed or Pb-free conditions. In addition, treatment with DDC (an inhibitor of SOD) and ATZ (an inhibitor of CAT) [62] further reduced the GR in Pb- and Pb+IDS-treated maize seeds after incubation for 24 h (Figure 6). This reduction confirmed that IDS can attenuate Pb stress-induced ROS overproduction to a certain extent. In contrast, treatment with the inhibitors of SOD and CAT enhanced the GR in distilled water- and IDS-treated maize seeds (Figure 6). These results suggest that IDS reduces ROS production in germinating seeds and ROS-scavenging enzymes (e.g., SOD and CAT) and also play a notable role in IDS-regulated seed germination under favorable and Pb-stressed conditions.

To further investigate the IDS-regulated ROS balance during maize seed germination under Pb stress, we measured the H_2_O_2_ content of seeds after the incubation period. Pb stress induced H_2_O_2_ overproduction in germinating seeds. However, IDS significantly decreased the Pb-enhanced H_2_O_2_ levels and reduced H_2_O_2_ accumulation under Pb-free conditions (Figure 7). NOX, a major ROS-producing enzyme, plays a key role in regulating seed germination [60]. Our results show that IDS treatment slightly (but significantly) reduced NOX activity compared with that by distilled water treatment (Figure 8). In contrast, Pb stress significantly enhanced NOX activity. However, Pb-enhanced NOX activity was significantly attenuated by IDS (Figure 8). Thus, the results of the addition of ROS-related redox agents and ROS-related enzyme activity assays can be partially attributed to the chelating activity of IDS, that is, IDS chelates Ca^2+^, restricting the activation of NOX [61]. These results also suggest that NOX-produced ROS are required for IDS-regulated seed germination.

The activities of ROS-scavenging enzymes (SOD and CAT) increased gradually during seed germination in maize. However, compared with those in the control, SOD and CAT activities decreased significantly in the IDS-treated groups after incubation for 1, 3, and 5 d (Figure 9 and Figure 10). This can be partly attributed to the decrease in IDS-induced NOX activity, which resulted in decreased levels of ROS, such as O_2_^−^ and H_2_O_2_. Pb stress increased SOD and CAT activities, which were further enhanced upon IDS treatment of the maize seeds after incubation. This finding indicates that IDS requires increased activities of ROS-scavenging enzymes (such as SOD and CAT) to improve seed germination under Pb stress. The single and combined treatment with IDS and Pb revealed that under Pb stress, IDS, as a chelating agent of metal ions, did not restrict the activities of metal-containing SOD and CAT by chelating metal ions, such as Fe, Cu, Mn, and Zn, which are required for the activation of these enzymes [41], but played a promoting role in comparison with the inhibitory effect of chelating Ca^2+^ on the activity of NOX. Therefore, further investigation is required to elucidate the mechanism underlying the regulation of antioxidant enzymes by IDS. However, protecting antioxidant enzymes from Pb-induced overproduction of ROS or free radicals, which can inactivate SOD and CAT [63,64], may be one of the crucial ways to promote seed germination and rescue seeds using IDS (Figure 2 and Figure 3).

To better understand the aforementioned findings, we propose a hypothetical model based on ROS levels to illustrate the mechanisms underlying IDS-regulated seed germination in Pb-free and Pb-stressed conditions (Figure 11). This model proposes that IDS-regulated seed germination in maize is closely associated with ROS levels. IDS reduces the activities of ROS-producing (e.g., NOX) and ROS-scavenging (e.g., SOD and CAT) enzymes in maize seeds incubated in normal environments. However, under conditions of Pb stress, IDS inhibits the activity of ROS-producing enzymes, increases the activities of ROS-scavenging enzymes, and promotes the germination of seeds in a Pb-stressed environment. In this above process, we suggest that Ca^2+^ chelation, which restricts NOX activity and protects and promotes SOD and CAT activity by inhibiting ROS overproduction, may be important for maintaining the ROS balance and improving seed germination and seedling growth under Pb stress resulting from IDS treatment in maize. In addition, IDS slightly interferes with the activity of SOD and CAT through the chelation of metal ions such as Fe, Zn, Cu, and Mn.

## 4. Materials and Methods

### 4.1. Reagent Preparation

All chemical reagents used in this work were of analytical grade. DPI, DMTU, DDC, and ATZ were purchased from Sigma-Aldrich (St. Louis, MO, USA). Other reagents were purchased from Aladdin Biochemical Technology Co., Ltd. (Shanghai, China).

### 4.2. Seed Treatment Groups

The seeds of the hybrid maize variety ‘Zhengdan 958’ used in the experiments were purchased from DONEED Seed Industry Co., Ltd, Beijing, China. The seeds were harvested in October 2020 and used in this experimental study in April 2021. The seed purity was greater than 99%, and the GR was greater than 95%. Before sowing, the seeds were surface-sterilized with 0.1% NaClO for 5 min. Healthy and uniform seeds with an intact seed coat were selected and submerged in distilled water (25 °C) for 1 h. The swollen seeds were then sown in a plastic box containing quartz stones (diameter: approximately 2–3 mm) to a depth of 2 cm. The plastic box was covered with parafilm and placed in an incubator (KWB720, Binder GmbH, Tuttlingen, Germany). After incubation for 1, 3, and 5 d, the germinated seeds were collected and stored at −80 °C for use in further analysis.

The experiment included three treatment groups (all at 25 °C) as follows:
Group 1: The swollen seeds were sown in a plastic box containing quartz stones moistened with 0, 5, 20, or 100 mmol·dm^−3^ IDS solution. The GR and seedling growth were monitored after sowing.Group 2: In accordance with the results of our previous study [45], we used 20 mmol·dm^−3^ PbCl_2_ as the Pb stress treatment condition. The swollen seeds were sown in a plastic box containing quartz stones moistened with 20 mmol·dm^−3^ PbCl_2_ + 0 mmol·dm^−3^ IDS, 20 mmol·dm^−3^ PbCl_2_ + 5 mmol·dm^−3^ IDS, 20 mmol·dm^−3^ PbCl_2_ + 20 mmol·dm^−3^ IDS, or 20 mmol·dm^−3^ PbCl_2_ + 100 mmol·dm^−3^ IDS solution. GR and seedling growth were monitored after sowing.Group 3: The swollen seeds were sown in a plastic box containing quartz stones moistened with distilled water (control), PbCl_2_ (20 mmol·dm^−3^), IDS (100 mmol·dm^−3^), or PbCl_2_ (20 mmol·dm^−3^) + IDS (100 mmol·dm^−3^) solution. Subsequently, DMTU (a specific scavenger of ROS, 10 mmol·dm^−3^), H_2_O_2_ (10 mmol·dm^−3^), DPI (a specific inhibitor of NOX, 0.1 mmol·dm^−3^), CaCl_2_ (0.1 mmol·dm^−3^), DDC (a specific inhibitor of SOD, 1 mmol·dm^−3^), and ATZ (a specific inhibitor of CAT, 1 mmol·dm^−3^) were applied separately. One milliliter of each ROS-related reagent solution was uniformly added to the quartz stones moistened with the distilled water, PbCl_2_, IDS, or PbCl_2_+IDS solutions, respectively.

All assays were repeated at least five times to minimize experimental error, and each replicate contained 100 seeds. The treatment concentration used in this study was based on the results of our preliminary experiments.

### 4.3. Determination of GR

The maize seeds were considered to have germinated when radicle emergence from the seeds was ≥1 mm. The number of germinated seeds was counted three times per day. The final GR was calculated as the percentage of seeds that had germinated 5 d after sowing [45].

### 4.4. Determination of Seedling Growth

Five-day-old maize seedlings (grown from seeds treated with PbCl_2_, IDS, or their combination) were washed with distilled water. Excess water was removed using bibulous paper, and the sprout length was measured using a vernier caliper.

### 4.5. Determination of H_2_O_2_ Content

The H_2_O_2_ contents of the maize seeds in different treatment groups (control, heavy metal-treated, and combination treatments) were determined using colorimetric measurements following the method described by Zhang et al. [65]. H_2_O_2_ was extracted by homogenizing germinating maize seeds (~0.5 g) in 3 mL of cold acetone. The samples were homogenized using a tissue homogenizer (Tissuelyser-24, Jingxin Industrial Development Co. Ltd., Shanghai, China) at 4 °C and immediately centrifuged at 12,000 rpm for 15 min. A 1 mL aliquot of the supernatant solution was mixed with 0.1 mL of 5% titanium sulfate in concentrated HCl, followed by the addition of 0.2 mL aqueous NH_3_ (25%) to precipitate the peroxide–titanium complex. The mixture was then centrifuged at 12,000 rpm for 15 min. The precipitate was solubilized in 3 mL of 2 mmol·dm^−3^ H_2_SO_4_, and the absorbance was measured at 415 nm. A standard response curve was prepared with a known concentration of H_2_O_2_ using the aforementioned method.

### 4.6. Determination of Antioxidant Enzyme Activities

Germinating maize seeds (~3 g) were ground to a powder in liquid nitrogen. The powder was then homogenized in 30 mL of an ice-cold extraction buffer, which consisted of 0.1 mol dm^−3^ Tris–HCl (pH 7.5), 0.23 mol dm^−3^ sucrose, 5% polyvinylpyrrolidone, 1 mmol·dm^−3^ EDTA, 10 mmol·dm^−3^ KCl, 10 mmol·dm^−3^ MgCl_2_, and 2.5 mmol·dm^−3^ ascorbic acid (added to the buffer just before use). After homogenization and centrifugation (12,000 rpm for 20 min), the supernatants were used for the enzyme assays.

To measure SOD activity, the reaction buffer (containing 13 mmol·dm^−3^ methionine, 75 μmol dm^−3^ nitro blue tetrazolium, 100 μmol dm^−3^ EDTA, and 2 μmol dm^−3^ riboflavin) was mixed with different volumes of enzyme extract in 50 mmol·dm^−3^ phosphate buffer (pH 7.8). The mixture was then exposed to light for 15 min. An increase in absorbance at 560 nm was observed due to formazan formation. One unit of SOD activity was defined as the amount of enzyme that inhibited nitro blue tetrazolium photoreduction by 50% [66].

To measure CAT activity, a reaction mixture was prepared in such way that 3 mL of it contained 150 mmol·dm^−3^ potassium phosphate buffer (pH 7.0), 15 mmol·dm^−3^ H_2_O_2_, and 50 μL enzyme extract. The reaction was initiated by adding 15 mmol·dm^−3^ H_2_O_2_. CAT activity was determined based on the consumption of H_2_O_2_ (E = 39.4 dm^3^·mmol^−1^·cm^−1^) at 240 nm for 3 min [67].

The soluble protein content was determined using the Bradford method [68], with bovine serum protein as a standard.

### 4.7. NADPH Oxidase Enzyme Assay

The NOX enzyme was extracted, and its activity was assayed using a Plant NOX assay kit (GMS50096.3 v.A, GenMed Scientifics Inc., Arlington, MA, USA), following the manufacturer’s instructions. The reaction was initiated by adding 0.2 mmol·dm^−3^ NADPH to a mixture containing 100 mmol·dm^−3^ Tris–HCl and 1 mmol·dm^−3^ EDTA. NOX activity was calculated by monitoring the decrease in absorbance at 340 nm, and the molar extinction coefficient used for NADPH was 6.22 × 10^3^ dm^3^·mmol^−1^·cm^−1^ at pH 8.0 [69].

### 4.8. Data Analysis

All experiments included five replicates, and the values represent the mean ± standard deviation (SD) of the five replicates. Statistical analyses were performed using analysis of variance (ANOVA) with SPSS software (version 19.0; SPSS Inc., Chicago, IL, USA). The differences between treatments were determined by Duncan’s new multiple range test and considered statistically significant at *p* < 0.05.

## 5. Conclusions

Several significant conclusions can be drawn from the results of this study. First, IDS (100 mmol·dm^−3^) markedly delayed maize seed germination in the absence of Pb stress. Second, the Pb-mediated inhibition of seed germination and seedling growth was significantly alleviated by IDS. Third, IDS significantly inhibited the activity of ROS-producing enzymes, reduced the level of H_2_O_2_ in maize seeds under normal conditions, and markedly weakened the activities of ROS-producing enzymes under Pb stress. In contrast, the Pb stress-induced increase in the activities of ROS-scavenging enzymes was significantly enhanced by IDS. Under Pb stress, NOX (activated by Ca^2+^), SOD, and CAT were involved in maintaining the IDS-regulated ROS balance and promoting seed germination. Thus, Ca^2+^ chelation and protection of antioxidant enzymes may be essential for the regulatory roles of IDS. Overall, our results suggest that IDS can be used as an inexpensive, biodegradable, and readily available reagent for widespread application in agriculture to improve seed germination in Pb-polluted soils.

## Figures and Tables

**Figure 1 plants-11-02487-f001:**
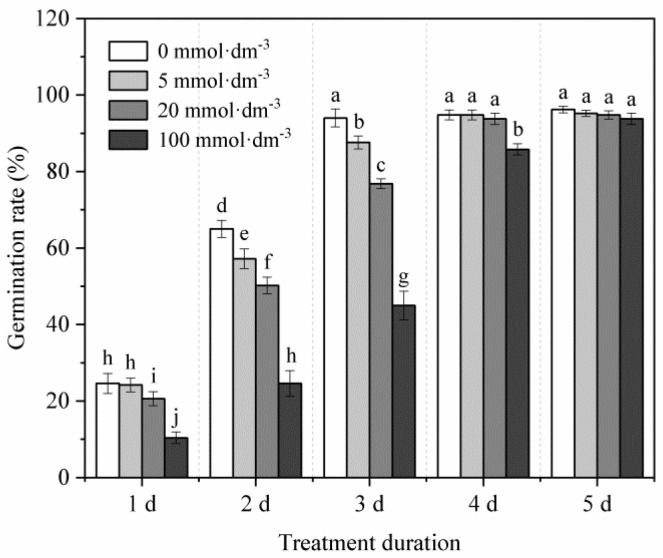
Effects of IDS on maize seed germination. Maize seeds were treated with different concentrations (0, 5, 20, and 100 mmol·dm^−^^3^) of the chelating agent IDS. The number of germinated maize seeds was recorded at 1, 2, 3, 4, and 5 d after sowing. Error bars represent the standard deviation of the mean (*n* = 5). Means associated with different letters are significantly different between treatments and time points (*p* < 0.05; Duncan’s new multiple range test). IDS, iminodisuccinic acid.

**Figure 2 plants-11-02487-f002:**
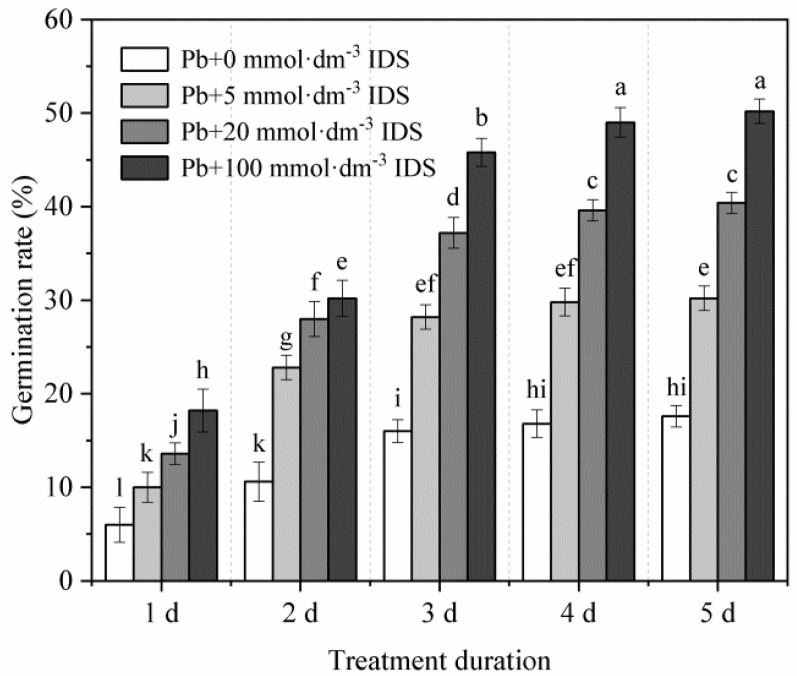
Effects of IDS on maize seed germination under Pb stress. Maize seeds were treated with 20 mmol·dm^−3^ PbCl_2_ + 0 mmol·dm^−3^ IDS, 20 mmol·dm^−3^ PbCl_2_ + 5 mmol·dm^−3^ IDS, 20 mmol·dm^−3^ PbCl_2_ + 20 mmol·dm^−3^ IDS, or 20 mmol·dm^−3^ PbCl_2_ + 100 mmol·dm^−3^ IDS solutions. The number of germinated maize seeds was recorded at 1, 2, 3, 4, and 5 d after sowing. Error bars represent the standard deviation of the mean (*n* = 5). Means associated with different letters are significantly different between treatments and time points (*p* < 0.05 level; Duncan’s new multiple range test). IDS, iminodisuccinic acid; Pb, lead.

**Figure 3 plants-11-02487-f003:**
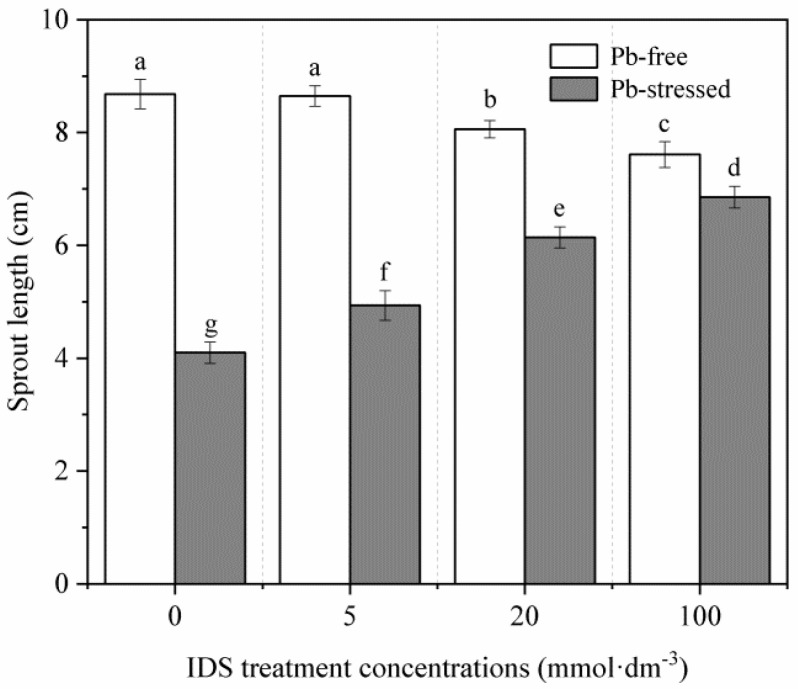
Effects of IDS on the sprout length of maize seedlings under Pb-stressed and Pb-free conditions. Maize seeds treated with Pb (20 mmol·dm^−3^ PbCl_2_) and distilled water were further treated with different concentrations (0, 5, 20, and 100 mmol·dm^−3^) of IDS solutions. The sprout lengths of 5-day-old maize seedlings were measured. Error bars represent the standard deviation of the mean (*n* = 5). Means associated with different letters are significantly different between treatments and time points (*p* < 0.05; Duncan’s new multiple range test). IDS, iminodisuccinic acid; Pb, lead.

**Figure 4 plants-11-02487-f004:**
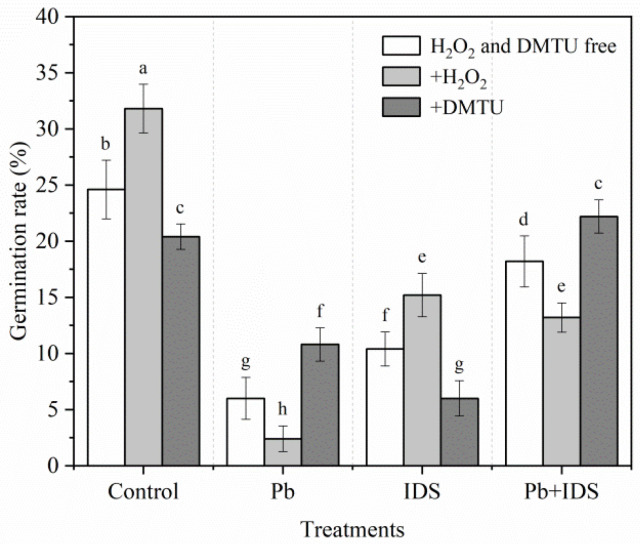
Effects of DMTU and H_2_O_2_ on the germination of maize seeds treated with IDS, Pb, or their combination. Maize seeds treated with distilled water (control), Pb (20 mmol·dm^−3^ PbCl_2_), IDS (100 mmol·dm^−3^), or Pb+IDS were further treated with DMTU or H_2_O_2_ for 1 d. Error bars represent the standard deviation of the mean (*n* = 5). Means associated with different letters are significantly different among treatments and time points (*p* < 0.05; Duncan’s new multiple range test). Pb, lead; IDS, iminodisuccinic acid; DMTU, dimethylthiourea.

**Figure 5 plants-11-02487-f005:**
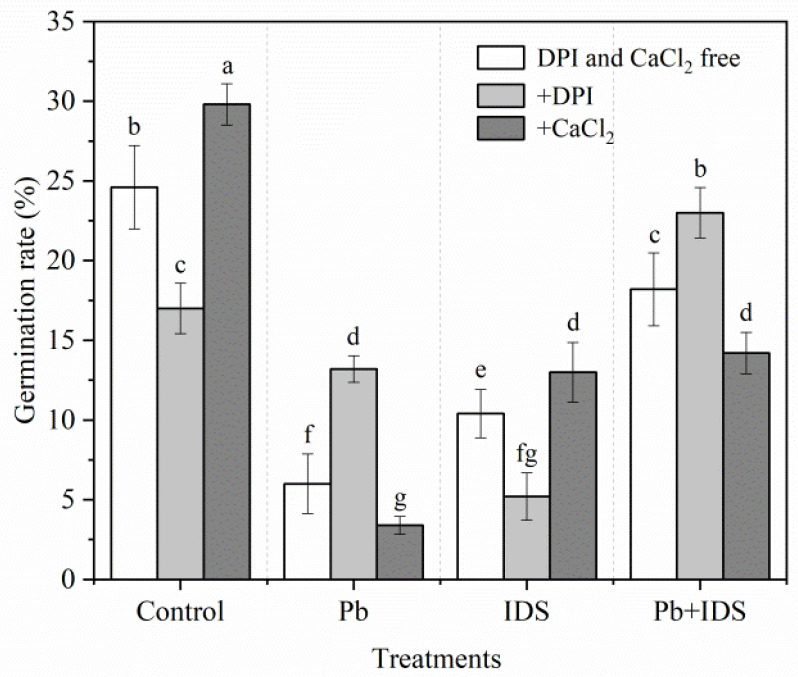
Effects of DPI and CaCl_2_ on the germination of maize seeds treated with IDS, Pb, or Pb+IDS. Maize seeds treated with distilled water (control), Pb (20 mmol·dm^−3^ PbCl_2_), IDS (100 mmol·dm^−3^), or Pb+IDS were further treated with DPI or CaCl_2_ for 1 d. Error bars represent the standard deviation of the mean (*n* = 5). Means associated with different letters are significantly different among treatments and time points (*p* < 0.05; Duncan’s new multiple range test). Pb, lead; IDS, iminodisuccinic acid; DPI, diphenyleneiodonium chloride.

**Figure 6 plants-11-02487-f006:**
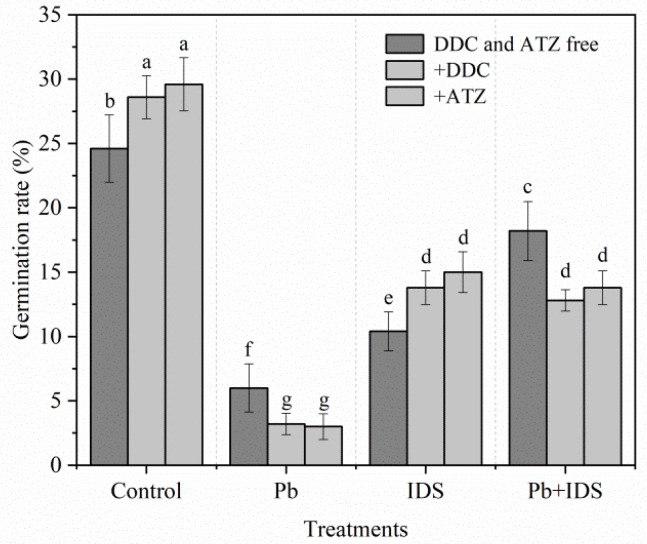
Effects of DDC and ATZ on the germination of maize seeds treated with distilled water, IDS, Pb, or Pb+IDS. Maize seeds treated with distilled water (control), Pb (20 mmol·dm^−3^ PbCl_2_), IDS (100 mmol·dm^−3^), or Pb+IDS were further incubated with DDC or ATZ for 1 d. Error bars represent the standard deviation of the mean (*n* = 5). Means associated with different letters are significantly different among treatments and time points (*p* < 0.05; Duncan’s new multiple range test). Pb, lead; IDS, iminodisuccinic acid; DDC, diethyldithiocarbamic acid; ATZ, aminotriazole.

**Figure 7 plants-11-02487-f007:**
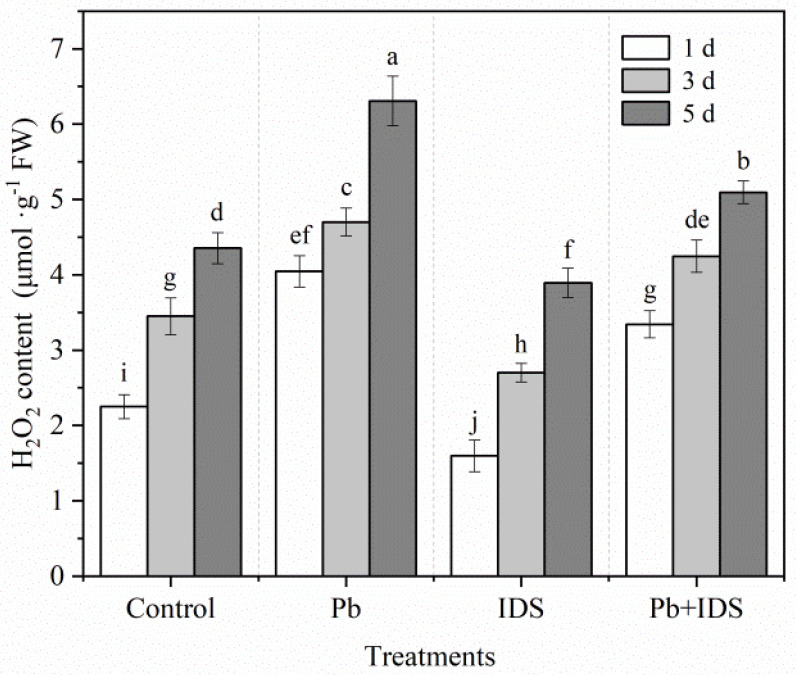
Effects of Pb, IDS, and Pb+IDS on the H_2_O_2_ content of maize seeds. Maize seeds treated with distilled water (control), Pb (20 mmol·dm^−3^ PbCl_2_), IDS (100 mmol·dm^−3^), or Pb+IDS were used, and their H_2_O_2_ contents were measured 1, 3, and 5 d after sowing. Error bars represent the standard deviation of the mean (*n* = 5). Means associated with different letters are significantly different among treatments and time points (*p* < 0.05; Duncan’s new multiple range test). Pb, lead; IDS, iminodisuccinic acid; FW, fresh weight.

**Figure 8 plants-11-02487-f008:**
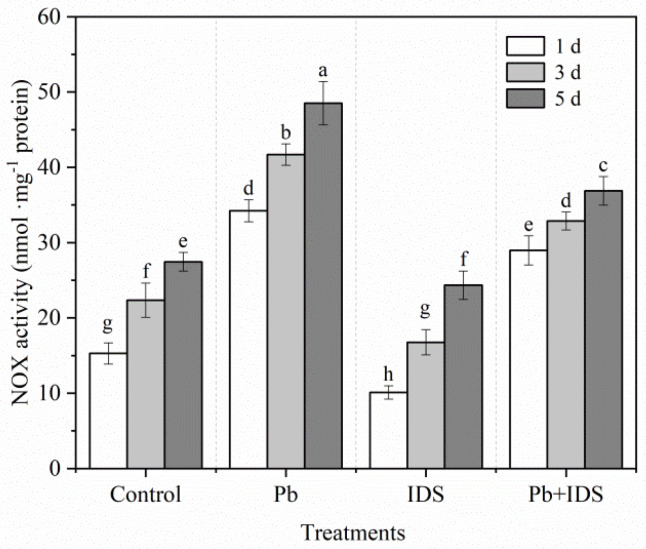
Effects of Pb, IDS, and Pb+IDS on NOX activities of maize seeds. Maize seeds incubated with distilled water (control), Pb (20 mmol·dm^−3^ PbCl_2_), IDS (100 mmol·dm^−3^), or Pb+IDS were used, and NOX activities were measured 1, 3, and 5 d after sowing. Error bars represent the standard deviation of the mean (*n* = 5). Means associated with different letters are significantly different among treatments and time points (*p* < 0.05; Duncan’s new multiple range test). Pb, lead; IDS, iminodisuccinic acid; NOX, NADPH oxidase.

**Figure 9 plants-11-02487-f009:**
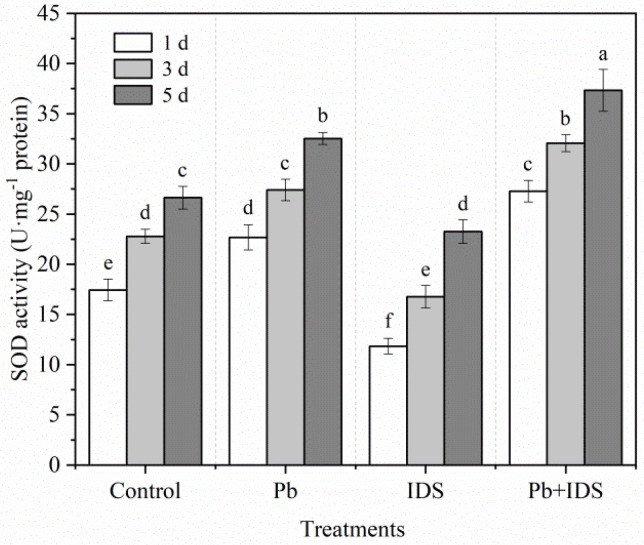
Effects of Pb, IDS, and Pb+IDS on SOD activities of maize seeds. Maize seeds treated with distilled water (control), Pb (20 mmol·dm^−3^ PbCl_2_), IDS (100 mmol·dm^−3^), or Pb+IDS were used, and their SOD activities were measured 1, 3, and 5 d after sowing. Error bars represent the standard deviation of the mean (*n* = 5). Means associated with different letters are significantly different among treatments and time points (*p* < 0.05; Duncan’s new multiple range test). Pb, lead; IDS, iminodisuccinic acid; SOD, superoxide dismutase.

**Figure 10 plants-11-02487-f010:**
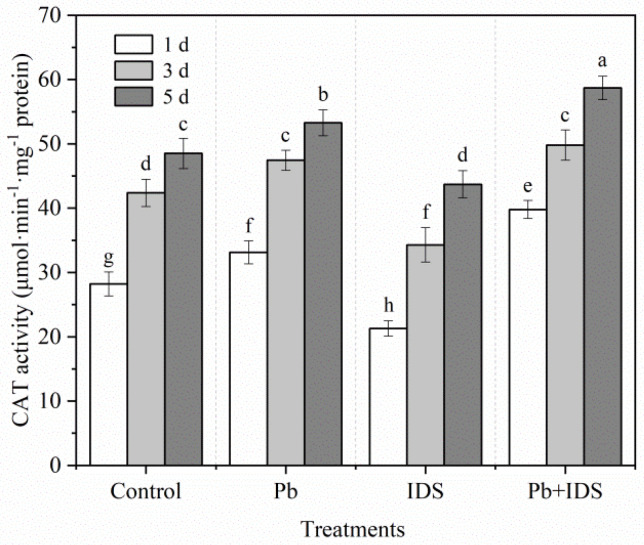
Effects of Pb, IDS, and Pb+IDS on CAT activities of maize seeds. Maize seeds treated with distilled water (control), Pb (20 mmol·dm^−3^ PbCl_2_), IDS (100 mmol·dm^−3^), or Pb+IDS were used, and their CAT activities were measured 1, 3, and 5 d after sowing. Error bars represent the standard deviation of the mean (*n* = 5). Means associated with different letters are significantly different among treatments and time points (*p* < 0.05; Duncan’s new multiple range test). Pb, lead; IDS, iminodisuccinic acid; CAT, catalase.

**Figure 11 plants-11-02487-f011:**
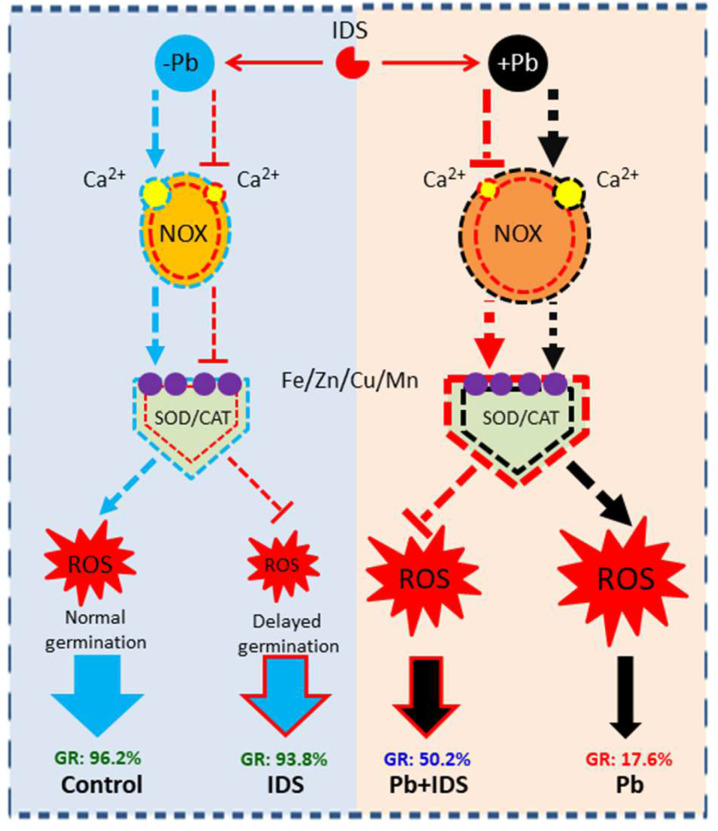
Hypothetical model based on ROS scavenging and production illustrating IDS-regulated seed germination processes under Pb-free (−Pb) and Pb-stressed (+Pb) conditions. The blue dotted line indicates the control treatment (distilled water), the black dotted line indicates treatment with Pb stress, and the red dotted line indicates treatment with IDS. Pointed and blunt dashed arrows represent the positive and negative effects of these treatments, respectively. The thickness of the dotted line indicates the degree of influence of different treatments on relevant indicator values. Pb, lead; ROS, reactive oxygen species; IDS, iminodisuccinic acid; NOX, NADPH oxidase; SOD, superoxide dismutase; CAT, catalase; GR, germination rate.

## Data Availability

All data generated or analyzed during this study are included in this published article.

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
