# Peer review of "Chelator Iminodisuccinic Acid Regulates Reactive Oxygen Species Accumulation and Improves Maize (Zea mays L.) Seed Germination under Pb Stress"

_plants, 2022, doi:10.3390/plants11192487_

Round 1

Reviewer 1 Report

I recommend the publication of the article because scientific experimentation on improving seed germination in polluted or abiotically stressed soil conditions is of particular interest.

The aim and objectives of the article have been stated and are very interesting. The use of new methodologies for plant propagation is an important topic especially under conditions of pollution or environmental stress. The work done is certainly of international interest and the format applied is certainly suitable for a research article. The work is original, of particular interest and can certainly stimulate research on this topic. The length of the article is appropriate for the journal and the graphs and tables are clear and easy to understand. The conclusion summarises the aims of the work and future prospects.

Reviewer 2 Report

General comments

The article is quite interesting. However, however it needs some small corrections

Specific comments:

Lines 114-115: what does 2.03- mean?. This needs to be explained better.

Figures 4, 5, 6, 7, 8, 9, 10. “Cont” is control?. This must be modified in all figures

Lines 162-180: this paragraph needs to be explained better

Figure 11: This figure is very complex. should be explained better
